# Fourier-Based Adaptive Signal Decomposition Method Applied to Fault Detection in Induction Motors

J. Jesus De Santiago-Perez, Martin Valtierra-Rodriguez , Juan Pablo Amezquita-Sanchez ,
Gerardo Israel Perez-Soto , Miguel Trejo-Hernandez and Jesus Rooney Rivera-Guillen *

CA Dynamic Systems and Control, Faculty of Engineering, Autonomous University of Queretaro,
San Juan del Rio 76010, Queretaro, Mexico
* Correspondence: jesus.rooney.rivera@uaq.mx; Tel.: +52-427-2741244

**Abstract:** Time-frequency analysis is commonly used for fault detection in induction motors. A variety of signal decomposition techniques have been proposed in the literature, such as Wavelet transform, Empirical Mode Decomposition (EMD), Multiple Signal Classification (MUSIC), among others. They have been successfully used in many works related with the topic. Nevertheless, the studied signals present amplitude changes and chirp-type frequency components that are difficult to track and isolate with the aforementioned techniques. The contribution of this work is the introduction of a novel technique for time-frequency signal decomposition that is based on an adaptive band-pass filter and the Short Time Fourier Transform (STFT), namely Fourier-Based Adaptive Signal Decomposition (FBASD) method. This method is capable of tracking and extracting nonstationary time-frequency components within a user-selected frequency band. With these components, a methodology for detecting and classifying broken rotor bars in induction motors using the startup transient current is also proposed.

**Keywords:** adaptive signal decomposition; Fourier; notch filter; fault detection; time-frequency analysis; broken rotor bar; induction motor

## 1. Introduction

Induction motors (IMs) have become a fundamental part of the industry since they are versatile electrical machines that are used in different areas of production processes [1–3]. The robustness, low cost, easy control and low maintenance requirements make the IMs highly relevant elements, representing 80% of the energy consumed by processes in the industry [4]. However, IMs, like any machine, are subject to faults which can be caused by alterations in operating conditions or by natural wear [5]. The most common problems of an IM appear in its main components: bearings, stator and rotor [6]. Regarding rotor failures, the presence of broken rotor bars (BRBs) is one of the main reported and studied problems in the literature. This failure occurs when a crack breaks one or more rotor bars, altering or interrupting the current flow through them [7]. Detection of the failure by BRBs in the IMs is a challenging problem to solve, especially when the failure occurs in an early state, i.e., when a bar is partially broken. In this condition, the IMs operate in an apparently normal way without visible alterations [8]. However, this faulty condition may continue and worsen over time, eventually impacting on the production line and affecting the operating costs [9]. For this reason, the development of techniques or methodologies that allow the detection of BRB faults in early states is an important task; moreover, correction and maintenance actions can be implemented in order to avoid major problems such as the machine collapse [10].

In recent years, the introduction of the Industry 4.0 and machine learning concepts in production processes has enabled the development of intelligent systems centered on the monitoring, storage and processing of data/signals. These systems are focused on the rapid

and reliable diagnosis of industrial machinery by detecting faults in a timely manner [11]. In particular, for the BRB fault detection, several proposals based on the analysis of different physical signals such as current, vibrations, ultrasound and temperature, among others have been developed, being the motor current signature analysis (MCSA), one of the most studied methodologies due to its easy monitoring, i.e., the signal can be acquired without the need to stop or alter the production processes [12–14]. Additionally, the MCSA stands out because its time-frequency representation shows specific components around the fundamental frequency associated with the BRB faults [15–17]. However, these frequency components are generally smaller in magnitude compared to the fundamental component and cannot be easily detected [13]. Therefore, there is a need for processing algorithms that can isolate the components associated with the BRB fault while separating or attenuating unwanted frequency components from the source signal [16].

In the last decades, a great variety of processing techniques have been reported in the literature, e.g., attenuation filters (notch filters) [18,19], FIR banks filters [20], Fast Fourier Transform [21–23], Welch's periodogram method [24], Wavelet transform [25,26], pattern recognition based on artificial neural networks [14] and recent signal processing techniques such as empirical mode decomposition (EMD) [27–29], Wavelet packet [30,31], and MUSIC [32,33], among others. The aforementioned techniques report good results in achieving their objective; however, there are still challenges that need to be addressed. For instance, digital filters such as notch filters and stop-band filters only focus on attenuating specific frequencies in a certain range and generally cannot have good results in the application of BRB faults detection in IM since the fundamental frequency that appears in MCSA does not always remain constant and can vary due to several factors [34] such as the quality of the power supply, the load connected to the electrical network or the same alteration due to motor faults. This fact also limits the performance of the Fast Fourier Transform (FFT) since this tool mainly deals with linear and stationary signals. An alternative to the FFT is the application of the Wavelet transform which separates a signal in different frequency bands; however, it is not automatic, since the mother wavelet and the levels of decomposition must be previously selected. Similarly, in recent years, EMD has been widely used for different applications where the decomposition of the signal is the main goal. This algorithm provides several components called intrinsic mode functions (IMFs). Although promising results have been obtained, it has the disadvantage that the frequency components to be isolated can get mixed in different IMFs, so the decomposition is not adequate; additionally, failures occur when the signal contains noise components. Subsequently, the ensemble empirical mode decomposition (EEMD) and the complete EEMD were developed as improvements to the original technique (i.e., EMD), whose objective is to reduce the above-mentioned disadvantages; however, these new techniques require a high computational load and do not fully combat the described problems. On the other hand, advanced processing techniques such as Wavelet Packet and MUSIC present good results for signal decomposition; however, they require a high computational load that limits them for developing applications in real time. Additionally, the knowledge of an expert is necessary to adjust and tune some initial parameters.

The contribution of this work is the introduction of the Fourier-Based Adaptive Signal Decomposition (FBASD) method for the analysis of nonlinear and nonstationary signals. The proposed technique combines the STFT and a second order band-pass digital filter to extract frequency components from a signal within a user selected frequency band. The effectiveness of the proposed technique is demonstrated by generating a FBASD-based methodology for automatically detecting and classifying BRB faults in IMs.

## 2. Materials and Methods

In this section, the proposed FBASD method is firstly described; then, a simulation-based study case that represents the BRB fault in IMs is presented and, finally, a comparison with the EMD and Wavelet methods is shown.

### 2.1. Fourier-Based Adaptive Signal Decomposition (FBASD)

For detecting and extracting frequency components that change in both amplitude and frequency, the new FBASD method shown in Figure 1 is developed. In general, it consists of two steps. Step 1 (Figure 1b): the STFT is used to determine the frequency response of a section of the input signal. This section has a window size represented by *WZ*. Then, the frequency component $fc_i$, with the maximum energy within an adaptive frequency range, $fc_{i-1} \pm D$, is obtained. *D* represents the frequency range. Once the $fc_i$ values are obtained, a continuous time-frequency trajectory, $f_k$, is computed through a first-order interpolation method. Step 2 (Figure 1c): A band-pass filter with adaptive cutting frequency, $f_k$, is used to extract the main time-frequency component, $E_k$, of the input signal; also, the residue signal, $R_k$, is obtained. A more detailed description for the abovementioned steps is presented in the next subsections.

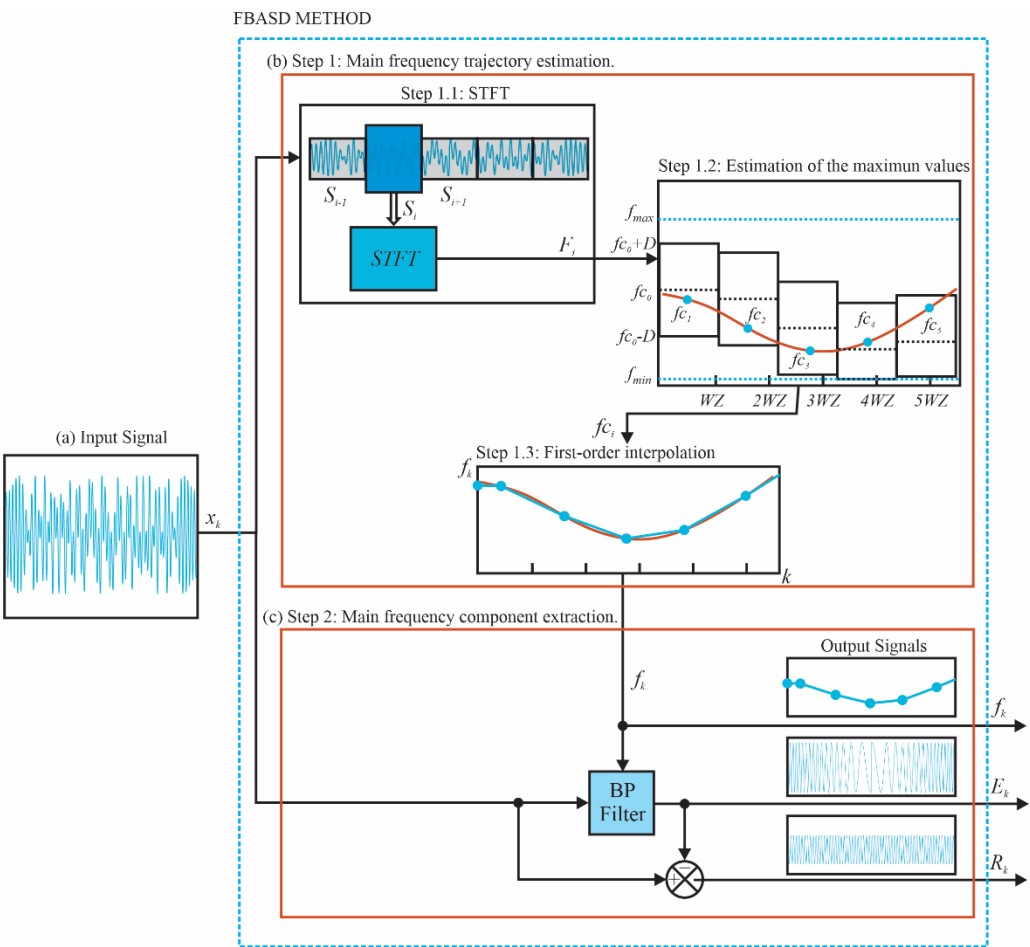

**Figure 1.** FBASD methodology. (**a**) Input signal, (**b**) Step 1: Main frequency trajectory estimation, (**c**) Step 2: Main frequency component extraction.

#### 2.1.1. Step 1. Main Frequency Trajectory Estimation

This step is used to obtain the trajectory of the frequency component with the biggest energy of an input signal.

Step 1.1. STFT: The discrete input signal, $x_k$, is segmented into a series of windows, $S_i$, of size *WZ*, i.e., $S_i \in [x_{i\,WZ}, x_{(i+1)\,WZ})$. For each window, the non-overlap STFT is computed for obtaining the spectra, $F_i$. The *WZ* is calculated as the number of samples

required for fitting a period of the minimum user-defined frequency, $f_{\min}$, to be tracked. This is achieved by means of Equation (1).

$$WZ = \left\lfloor \frac{F_S}{f_{\min}} \right\rfloor \tag{1}$$

where $F_S$ is the sampling frequency and $f_{\min}$ is the minimum frequency to be analyzed, which is defined by the user. $\lfloor \cdot \rfloor$ rounds down to the nearest integer number.

To compute the STFT, the discrete Fourier transform is used in this methodology according to Equation (2).

$$F_{i,m} = \sum_{n=0}^{WZ-1} x_{n+i\,(WZ)}\, e^{-\frac{2\pi j}{Fs} m\, n} \tag{2}$$

where $m = f_{\min}, f_{\min} + 1, \ldots, f_{\max}$.

For implementation purposes, the FFT algorithm is used instead and only the user-defined frequency range components $[f_{\min}, f_{\max}]$ are considered. This is an advantage over other methodologies since the user can select a specific region of interest to be analyzed, opening the use of the proposed method in different applications. Such frequencies are considered in this methodology to be positive integers and are selected by the user.

Step 1.2. Estimation of the maximum values. In this stage, the frequency of the main component, $fc_i$, within $WZ$ is computed by considering the maximum user-defined frequency rate of change, $froc$, which is given in Hz/s. This input parameter helps to stablish a frequency window, $D$, in which the main frequency component can change. This is achieved by means of Equation (3).

$$D = \left\lfloor froc\frac{WZ}{F_s} \right\rfloor \tag{3}$$

Therefore, the main frequency component, $fc_i$, is constrained within the range of $fc_{i-1} \pm D$. In this range, the frequency of the main component, $fc_i$, is simply the component with the biggest energy.

$$fc_i = \max|F_i| \tag{4}$$

With this approximation only integer values are obtained. As will be demonstrated in the study cases, this approach is accurate enough for the application given in this work. If a better estimation is required, the algorithm to obtain the main frequency component should be modified.

Step 1.3 First-order interpolation. Having as input the points $(i\,WZ,\ fc_i)$, a linear interpolation is used to obtain the Equation (5) which describes the time-frequency trajectory of $fc_i$.

$$f_k = \frac{f_{C_r} - f_{C_{r-1}}}{WZ}(k - r\,WZ) + f_{C_r} \tag{5}$$

where $r$ is calculated by means of Equation (6).

$$r = \left\lfloor \frac{k}{WZ} \right\rfloor + 1 \tag{6}$$

$k$ takes integer values of $\{0, 1, 2, 3, \ldots, K-1\}$. $K$ is the number of samples of the input signal.

### 2.1.2. Step 2. Main Frequency Component Extraction

This step considers the frequency trajectory, $f_k$, of the main component and extracts it from the input signal, $x_k$. This action is performed by a second-order band-pass filter. Thus, the input signal is decomposed into an extracted component, $E_k$, and a residue signal, $R_k$. The second order band-pass filter is defined by means of Equation (7).

$$G(s) = \frac{Bs}{s^2 + Bs + w_k^2} \tag{7}$$

where $B$ is the filter bandwidth which is calculated by means of Equation (8).

$$B = 2D \tag{8}$$

Furthermore, $w_k$ is the cutting frequency in rad/s and is calculated using Equation (9):

$$w_k = 2\pi f_k \tag{9}$$

The frequency response of the band-pass filter is shown in Figure 2. As can be observed in the cutting frequency, $w_k$, the magnitude is 0 dB and the phase is 0 degrees. Therefore, it is possible to subtract the estimated frequency component, $E_k$, from the input signal, $x_k$, in order to obtain the residue signal, $R_k$.

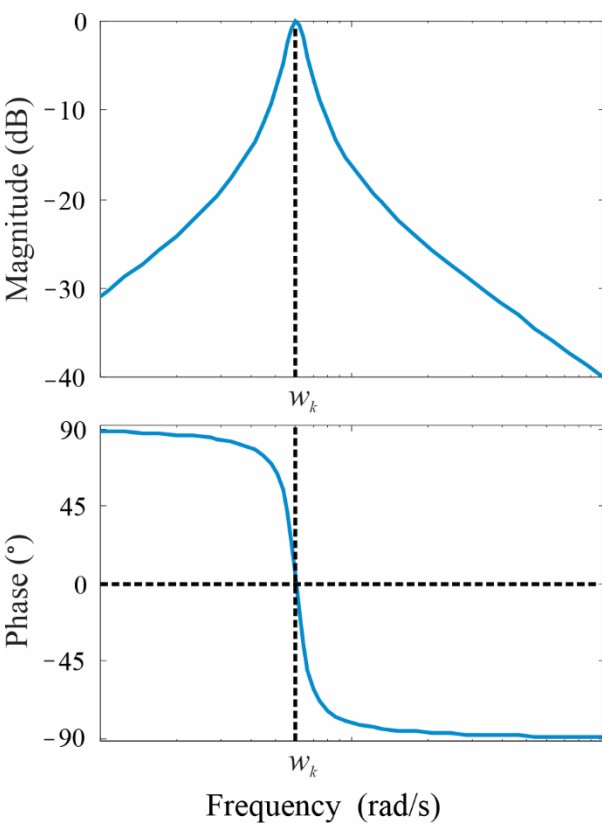

**Figure 2.** Filter response of the band-pass Filter.

The selected BP filter is digitalized by means of the bilinear transform [35]. The digital version of the transfer function is shown in Equation (10).

$$G(z) = \frac{\frac{2T_s B}{y} - \frac{2T_s B}{y} Z^{-2}}{1 + \frac{2w_k^2 - 8}{y} Z^{-1} + \frac{4 - 2T_s B + w_k^2}{y} Z^{-2}} \tag{10}$$

where $y = 4 + 2T_s B + w_k^2$ and $T_s$ is the sampling frequency of the digitalization process.

The bilinear transform approach affects the position of the cutting frequency. Then, it is necessary to correct its value by means of Equation (11).

$$w_k^* = \frac{2}{T_s} \tan\left(\frac{T_s}{2} w_k\right) \tag{11}$$

Finally, the difference equation required to implement the Digital BP filter is represented by means of Equation (12).

$$E_k = -\frac{2w_k^{*2} - 8}{y}E_{k-1} - \frac{4 - 2T_sB + w_k^{*2}}{y}E_{k-2} + \frac{2T_sB}{y}x_k - \frac{2T_sB}{y}x_{k-2} \tag{12}$$

### 2.2. Extraction of N Time-Frequency Components

The outputs of the FBASD method are three discrete-time-based vectors, i.e., the frequency trajectory, $f_k$, the extracted component, $E_k$, and a residue signal, $R_k$. If several components are required to be extracted, the residue signal becomes the new input signal and the two steps of the proposed FBASD technique are computed again. A block diagram depicting this process is shown in Figure 3. It shows the FBASD use for extracting n-time-frequency components.

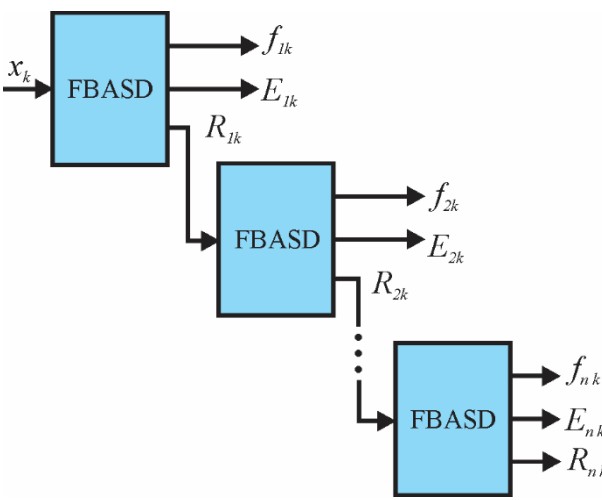

**Figure 3.** FBASD technique applied to extract N time-frequency components.

Summarizing, the proposed FBASD method is capable of tracking and extracting static and frequency-changing components from an input signal. The user is able to constrain the frequency range, where the frequency components are required being extracted. The method is also able to track specific frequency components by approximately specifying the maximum frequency rate of change of the components being extracted. The mentioned features are desirable for the fault detection of BRB in induction motors.

### 2.3. Simulation

The simulation consists of applying the FBASD technique to a synthetic signal emulating the startup current signal from an induction motor under BRB fault condition. It is well-known that, in this type of fault, sideband frequency components around the fundamental component (supply frequency) appear. In particular, the left sideband frequency component exhibits a V-shaped pattern in the time-frequency plane [36]; furthermore, harmonic components from different sources can be present. In this regard, the proposed synthetic signal has the following functions:

$$y(t) = g_1(t) + g_2(t) + g_3(t) + g_4(t) \tag{13}$$

where

$$g_1(t) = 0.3\sin\left(2\pi\left[f_a t + (f_b - f_a)(t-1)^3/3\right]\right)$$
$$g_2(t) = 3\sin(2\pi f_c t)$$
$$g_3(t) = \cos(2\pi f_d t)$$
$$g_4(t) = 0.4\cos(2\pi f_e t)$$

These components are shown in Figure 4. It consists of a concave quadratic chirp signal changing within the range of $f_a = 5$ to $f_b = 60$ Hz, emulating a frequency changing component associated with a BRB condition, i.e., the V-shaped pattern. Also, three sine signals are added with the following frequencies $f_c = 60$, $f_d = 120$ and $f_e = 180$ Hz, such components represent the fundamental component, second harmonic, and third harmonic, respectively. The composed signal has duration of 2 s and it is sampled at a frequency of 1500 samples/s. On the other hand, Wavelet and EMD methods are applied to the signal in order to compare their results with the proposed technique.

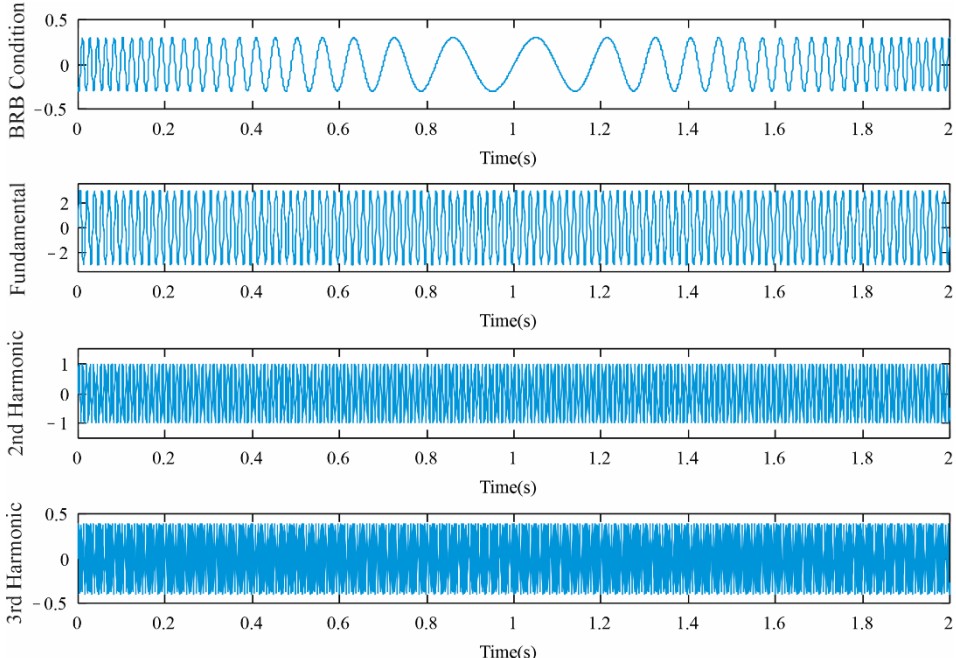

**Figure 4.** Components of the generated synthetic signal.

The configuration parameters of the proposed methodology are selected as follows: a frequency range of [4, 120] Hz, such frequency range includes the range where the frequency-moving component associated with the BRB fault occurs; however, for other applications the frequency range and, consequently, the filter must be adapted to include the harmonic component of interest. In this case, in order to show that other components that can be present in the input signal do not compromise the proposal performance, the third harmonic of the fundamental frequency is intended to be discriminated since it is outside the selected frequency range. A frequency rate of change of $froc = 60$ Hz/s is chosen since the chirp-type signal that is required to be extracted changes at a similar rate. The number of components to be extracted from the input signal is set to 3. Nevertheless, this number can be modified to extract more or less components as the user requires. If the user selects more frequency components than the existing in the input signal, the method will output noise in the additional components.

The simulation results are shown in Figure 5. The first row shows the input signal which is the sum of all the components shown in Figure 4. The second row shows the FBASD outputs for the first iteration. It consists of the residue signal, $R_1$, the frequency tracking, $f_1$, and the extracted component, $E_1$. This extracted component corresponds to the fundamental component, $g_2(t)$. The third row is composed of the residue signal, $R_2$, the frequency tracking, $f_2$, and the extracted component, $E_2$. In this case, the extracted component is the first harmonic component, $g_3(t)$. Finally, the fourth row is composed of a residue signal, $R_3$, a frequency tracking, $f_3$, and the extracted component, $E_3$. In this iteration, the extracted component matches the chirp-type component, $g_1(t)$, which emulates the BRB condition. It is worth noting that the proposal first extracts the components with

the bigger amplitude and the smallest ones are detected at the end. The component $g_4(t)$ is discriminated by the proposed FBASD method since it oscillates at a frequency of 180 Hz which is outside of the user-defined frequency range, i.e., [4, 120] Hz.

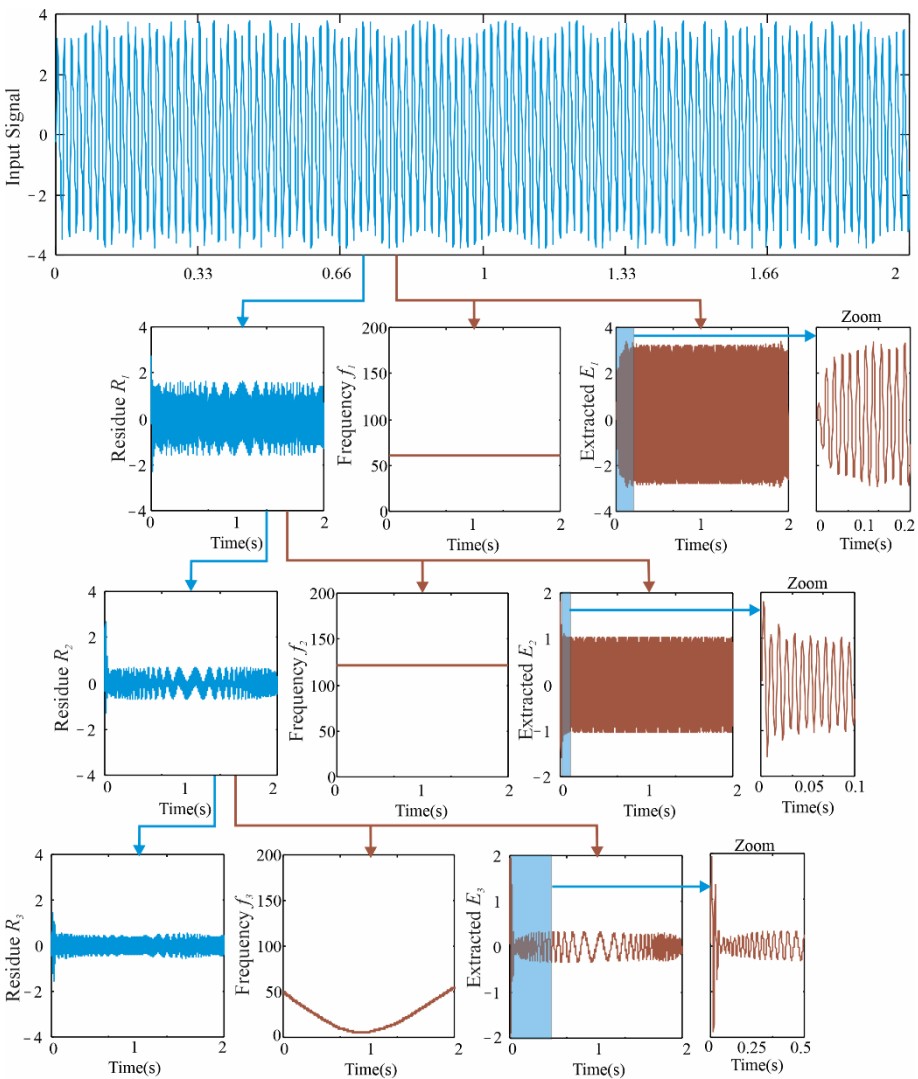

**Figure 5.** Simulation results for the extraction of three frequency components.

### 2.4. Comparison

The proposed FBASD technique is compared with the EMD and Discrete Wavelet transform (DWT). In particular, the EMD method is characterized by being an adaptive method with the capability of decomposing a time signal in different frequency bands or intrinsic mode functions (IMFs) according to its frequency information [37]. On the other hand, DWT provides a time-scale analysis of the in-test signal. It is characterized by decomposing a signal in different frequency bands called approximations (discrete-time low-pass filters) and details (discrete-time high-pass filters), respectively, where the approximation obtained in the first level is divided into a new approximation and a new detail [38]. This process is repeated according to the number of decomposition levels.

For the comparison, the EMD, featuring spline interpolation [37], and the DWT are applied to the Equation (13). For the Wavelet decomposition, a Discrete Meyer (DMEY) mother wavelet and four decomposition levels are selected since this configuration provide suitable results for the specific synthetic signal [26]. The outputs of both EMD and DWT techniques are shown in Figure 6. The fundamental component, $g_2(t)$, is observed in the IMF1 plot for the case of the EMD decomposition and in the detail level d4 for the case

of the Wavelet decomposition. The chirp-type signal, $g_1(t)$, that corresponds to the BRB condition is found in IMF2 for the case of EMD and in approximation level a4 for the case of the Wavelet decomposition. The second harmonic component, $g_3(t)$, is found in the decomposition level d3 of the Wavelet decomposition. The third harmonic component, $g_4(t)$, is found in the decomposition level d2. IMF3 and IMF4 do not give useful information about the input signal.

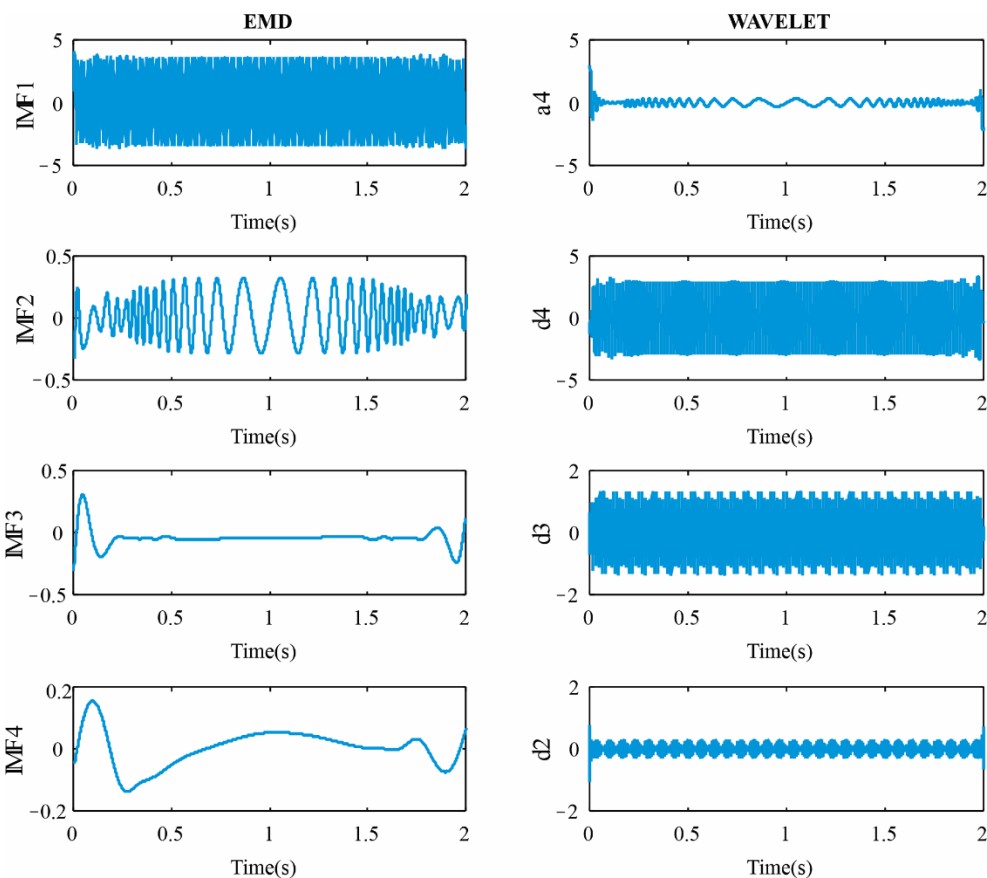

**Figure 6.** Signal decomposition using EMD and Wavelet.

The simulation is repeated adding 10 dB of white Gaussian noise to the synthetic signal as can be seen in Figure 7 in order to observe the noise impact.

The outputs of the FBASD, EMD, and Wavelet decomposition techniques are shown in Figure 8. As can be observed, the chirp-type signal, $g_1(t)$, that corresponds to the BRB condition is found in both the extracted $E_3$ component of the proposed FBASD and the a4-level approximation of the Wavelet decomposition. The EMD partially gives information about the BRB condition in IMF4 and IMF5 components. Additionally, the proposed FBASD technique outputs the frequency tracking, $f_3$, of the extracted component. The fundamental frequency component is found in $E_1$ with a frequency tracking given by $f_1$ from the FBASD technique. The Wavelet detects this component in the d4-level decomposition. The EMD gives mixed information about the fundamental frequency in IMF2 and IMF3, which is a common problem in this technique, i.e., mode-mixing. The second harmonic is visible in $E_2$ and $f_2$ from the FBASD. This component is also found in both the IMF1 from the EMD and d3-level from the Wavelet. The third harmonic is discriminated from the FBASD technique since such a frequency component is outside the selected frequency range. The EMD contains some information of such a component in the IMF1, whereas the Wavelet decomposition obtains it in the d1- and d2-levels.

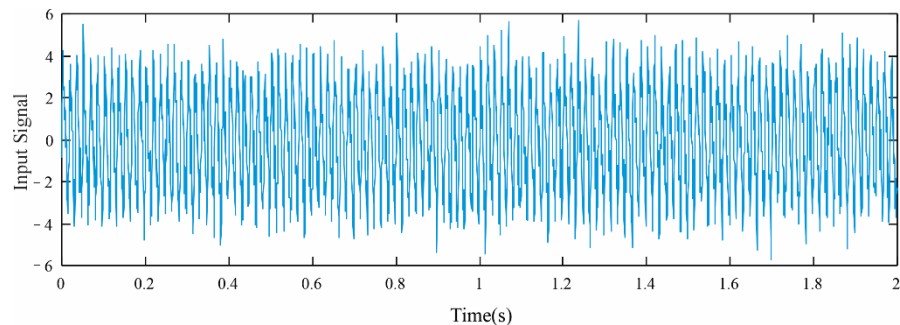

**Figure 7.** Synthetic signal with 10 dB of white Gaussian noise.

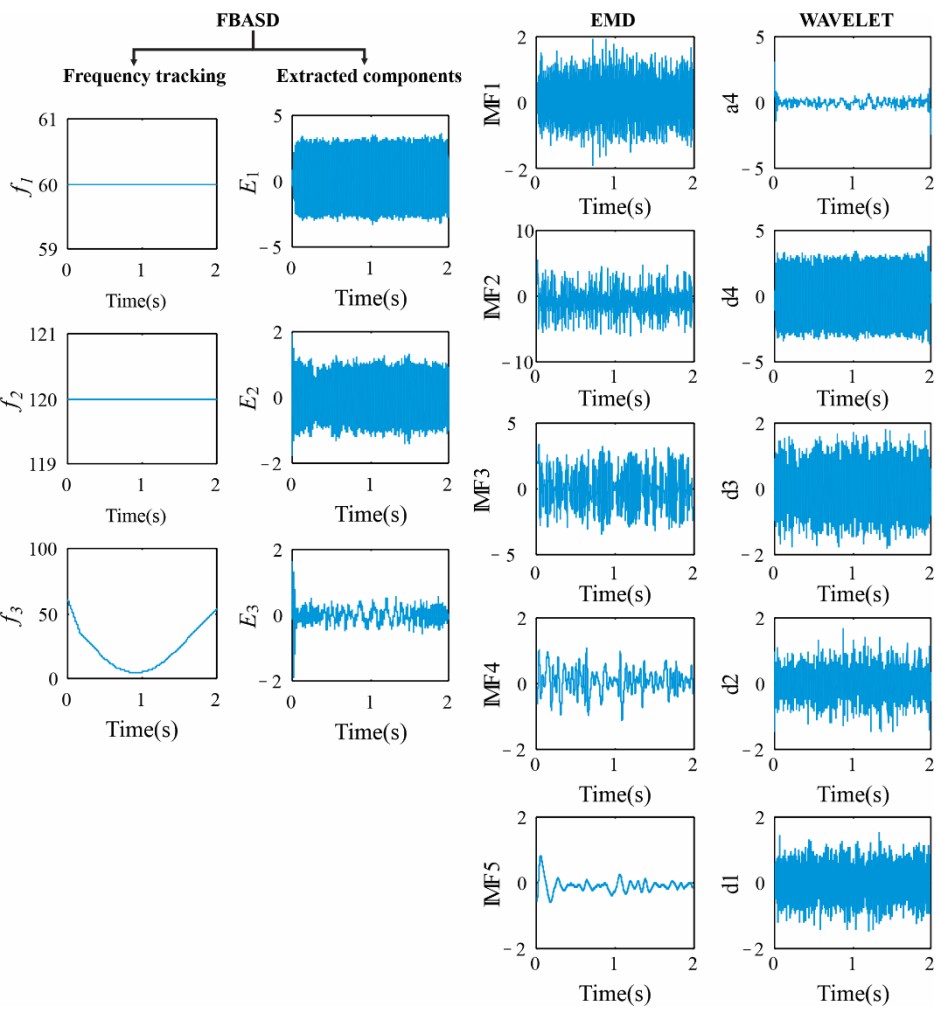

**Figure 8.** Simulation results with added noise.

In order to quantify the pertinence of the obtained results, a numerical comparison between the ideal signals (Equation (13)) and the extracted signals using the FBASD, EMD, and Wavelet methods, as well as the normalized mean square error (NMSE) is carried out for both ideal and noisy synthetic signals. The obtained results are shown in Table 1. The NMSE is a value less than or equal to 1, where 1 represents a high similarity between the signals. For the noiseless synthetic signal, the FBASD was capable of extracting the fundamental component as good as the Wavelet decomposition method since the two methods achieve values close to 1. For the signal that emulates the BRB condition, the best performance is obtained by the Wavelet, followed by the proposed FBASD, and the EMD last. The second harmonic was only detected by the Wavelet and FBASD methods, where

the latter method gave the best results. Results from the processing of the noisy signal show that the FBASD and Wavelet decomposition have the same performance for extracting the fundamental component. The EMD gave poor results with this component. The frequency component associated to the BRB condition was detected by all the techniques; however, the FBASD gave the best results. The second harmonic component was extracted by the Wavelet and the FBASD decompositions, being the FBASD the one with the best result. Finally, it is important to mention that, though the Wavelet method presents slightly better results than the ones obtained by the proposed method in noiseless conditions, the proposal is demonstrated to be more robust in noisy condition, which is a very important characteristic since in practice the noise is unavoidable. In addition, the Wavelet method requires an appropriate selection or configuration of two relevant parameters known as mother wavelet and decomposition level, which can affect the obtained results if they are not chosen adequately [7]. On the contrary, the proposed method can be considered as a proposal of low computational complexity, and it does not require a complex configuration.

**Table 1.** Quantitative comparative using NMSE.

| Signal | Method | NMSE Ideal | NMSE Noisy |
|---|---|---|---|
| | FBASD | 0.989 | 0.980 |
| Fundamental $g_2(t)$ | EMD | 0.867 | 0.364 IMF2 |
| | Wavelet | 0.987 | 0.980 |
| | FBASD | 0.830 | 0.340 |
| BRB Condition $g_1(t)$ | EMD | 0.648 | 0.207 IMF4 |
| | Wavelet | 0.900 | 0.250 |
| | FBASD | 0.950 | 0.900 |
| Second harmonic $g_3(t)$ | EMD | N.D. * | N.D. * |
| | Wavelet | 0.880 | 0.790 |

* Not detected.

On the other hand, a qualitative comparison between the FBASD, EMD, and Wavelet methods is also carried out (see Table 2). As can be observed, the proposed FBASD is a semi-adaptive technique that requires the setting of a frequency range and a maximum frequency rate of change. The last parameter helps to avoid mode mixing which is a very common problem in the EMD technique. The Wavelet is a non-adaptive technique that requires the selection of adequate Wavelet mother and decomposition level to give useful results. Unlike the proposal, EMD and Wavelet methods cannot provide frequency information, which can be useful for different applications. Also, in the proposed method, the frequency range of interest can be selected, which allows focusing the analysis and obtain somehow better results.

**Table 2.** Features for the FBASD, EMD, and Wavelet methods.

| Feature | FBASD | EMD | Wavelet |
|---|---|---|---|
| Basis | Semi-adaptive | Adaptive | Non-adaptive |
| Configuration parameters | Frequency range, *froc* | Standard deviation | Wavelet Mother, decomposition level |
| Frequency information | Yes | No | No |
| Frequency range of interest | Yes | No | Partially |
| Computing time (seconds) | 0.005 | 1.83 | 0.023 |

Regarding computational burden, Table 2 in the last line summarizes the mean computing time of the proposed algorithm (FBASD), EMD, and DWT for decomposing in individual frequency bands the synthetic signal on a personal computer MSI, featuring an intel core i7—10th generation which runs at 2.6 GHz and 40 GB RAM. According to the obtained results, it is important to mention that the FBASD is demonstrated to be of low complexity since it requires a low quantity of computational resources; for instance, its

computational cost is from 5 to 300 times less than the one obtained by DWT and EMD, respectively. All the methods were implemented in MATLAB software.

From the obtained results, it is demonstrated the capability of the proposed FBASD technique for decomposing time-varying signals, especially the noisy chirp-type components where the proposal outperforms the EMD and the Wavelet techniques. Therefore, the proposed FBASD method is a great candidate for detecting and isolating frequency changing components, such as the ones presented in the issue of BRB fault detection.

## 3. Results

The proposed FBASD is used to determine the BRB condition in induction motors using MCSA. In order to do so, the current signal during the startup transient of an induction motor under different rotor conditions is processed; next, a classification algorithm is proposed to differentiate between healthy, half broken bar, one broken bar, and two broken bar conditions.

### 3.1. Experimental Setup

After observing the obtained results from the simulation stage, a test system was developed to validate the performance of the proposed methodology with real current signals. The experimental setup is presented in Figure 9a. The three-phase induction motor model WEG-00136APE48T was used, having a nominal power of one hp, two poles, and 28 bars. The system was connected to a power supply of 220 Vac at 60 Hz. The mechanical load was applied by means of a four-quadrant dynamometer model 8540 from LabVolt. For data acquisition, the current signal from one phase of the induction motor was monitored with a current clamp model i200 from Fluke Corporation, where the data acquisition system (DAS) was based on the NI-USB 6211 from National Instruments. The sampling frequency was set at 1500 samples/s, where 20 tests for each condition were carried out for statistical purposes. Four rotor conditions were analyzed in the experiment: healthy (HLT), incipient failure of half broken bar (HBRB), and consolidated faults with one (1BRB) and two (2BRB) broken bars. For this purpose, the rotor bar faults were artificially induced by drilling a half bar hole, one bar, and two adjacent bars, respectively. The rotors with the induced damage are presented in Figure 9b. The overall methodology was implemented on MATLAB (version 9.11 R2021b from Mathworks, Inc., Natick, MA, USA).

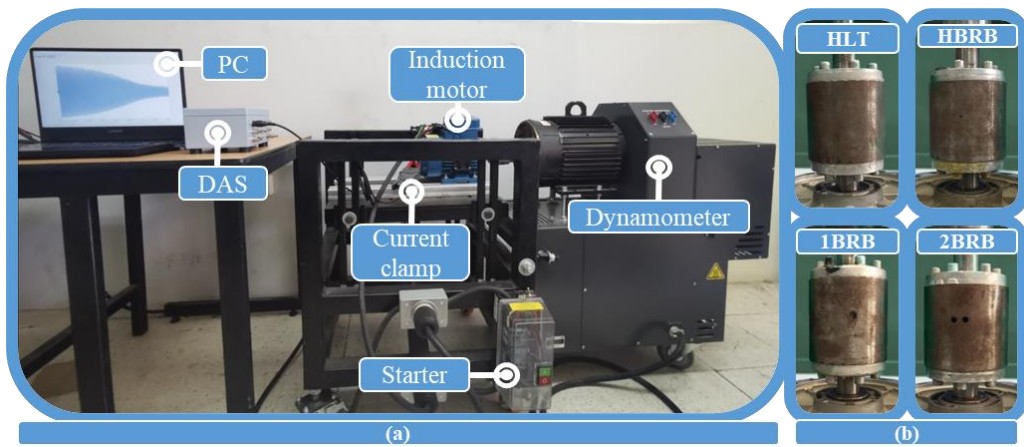

**Figure 9.** (**a**) Experimental setup and (**b**) Rotor conditions.

### 3.2. FBASD Results

The proposed FBASD method requires being configured for the specific application. Since the proposed algorithm performs the tracking and extraction of the frequency-moving components based on the STFT, adequate *WZ* is required. A big *WZ* increases the frequency accuracy, but at expenses of degrading the accuracy in the time domain. For the analysis, a frequency range of [4, 90] Hz is selected, since the component associated with the BRB

can move within this range in the startup transient of the motor [39]. With the sampling rate and the minimum frequency of $f_{min} = 4\text{Hz}$ a *WZ* of 375 samples is obtained. Next, adequate frequency rate of change *froc* is needed to be proposed. If no information is available, it can be considered the time duration of the experiment *T* and the frequency range desired being tracked for establishing a minimum *froc*. In this case, this value is $(f_{max} - f_{min})/T = (90 - 4)/2.7 = 31.85$ Hz/s. This means that the obtained value would be adequate for tracking a frequency component changing linearly within the desired range. Since frequency component associated with the BRB condition performs a V-shape during the current startup transient, three times the obtained value is proposed. For this reason, a *froc* = 100 Hz/s is chosen. Finally, the FBASD is configured to extract three time-frequency components from the current signal within the user-defined frequency range. This parameter is selected by considering that, in the case of 2BRB, three components of interest are present in the signal, lower V-shape component, 60 Hz component and higher V-shape frequency component.

The obtained results using the FBASD are shown in Figures 10–13 for the motor conditions of HLT, HBRB, 1BRB, and 2BRB, respectively. In Figure 10, the decomposition of the current signal for the healthy case is shown. The first row shows the motor current during the startup. The next three rows show the FBASD output for each iteration. In the first iteration, the 60 Hz frequency component, $E_1$, is extracted. The remaining signals will be compared with the ones obtained for the other IM conditions in order to assess differences and stablish an automatic classification algorithm. Figure 11 shows the FBASD output for the case of HBRB. As can be observed, $E_1$ contains the information of the 60 Hz component. Next, $E_2$ contains the information of the frequency changing component associated with this faulty case since its frequency tracking depicts somehow the left sideband frequency component associated to the BRB condition, where it is worth noting that the V-shaped pattern in the transient state is almost unperceivable because the fault is HBRB, i.e., an incipient fault that slightly affects the current signal; yet, in the steady state the left side band frequency component is present. $E_3$ does not contain useful information. Similarly, Figure 12 shows the results for the case of 1BRB. Again, $E_2$ contains the time-frequency information of the faulty case, where the V-shaped pattern is more evident. Finally, the results for the 2BRB are shown in Figure 13. In this case, the V-shaped frequency changing component is clearly detected in $f_2$ and extracted in $E_2$.

In all the study cases, where a fault is induced to the motor, the FBASD was capable of detecting and isolating the associated time-frequency component in the second iteration. Therefore, useful information is available in the $f_2$ and $E_2$ components, which can be used for proposing a classification algorithm.

### 3.3. Classification Algorithm

Once the proposed decomposition algorithm has obtained the main components of the measured current signal from the IM, the next step is to develop a stage of signal classification. This stage consists of analyzing the components extracted in order to automatically determine the IM condition (HLT, HBRB, 1BRB, and 2BRB). The components extracted from the signal correspond to the tracking of the main frequencies present in the current signal, since the first component is related to the fundamental frequency of the power supply (around 60 Hz) [13], the classification analysis will focus on the second extracted component that corresponds to the IM fault. In previous works, it has been reported that the spectrogram of a current signal shows that the amplitude of the frequencies associated with fault broken rotor bars increases according to the degree of damage [40]. Consequently, the amplitude of the second decomposition component, which is associated with the motor failure component, also should gradually increases according to the degree of severity in the IM.

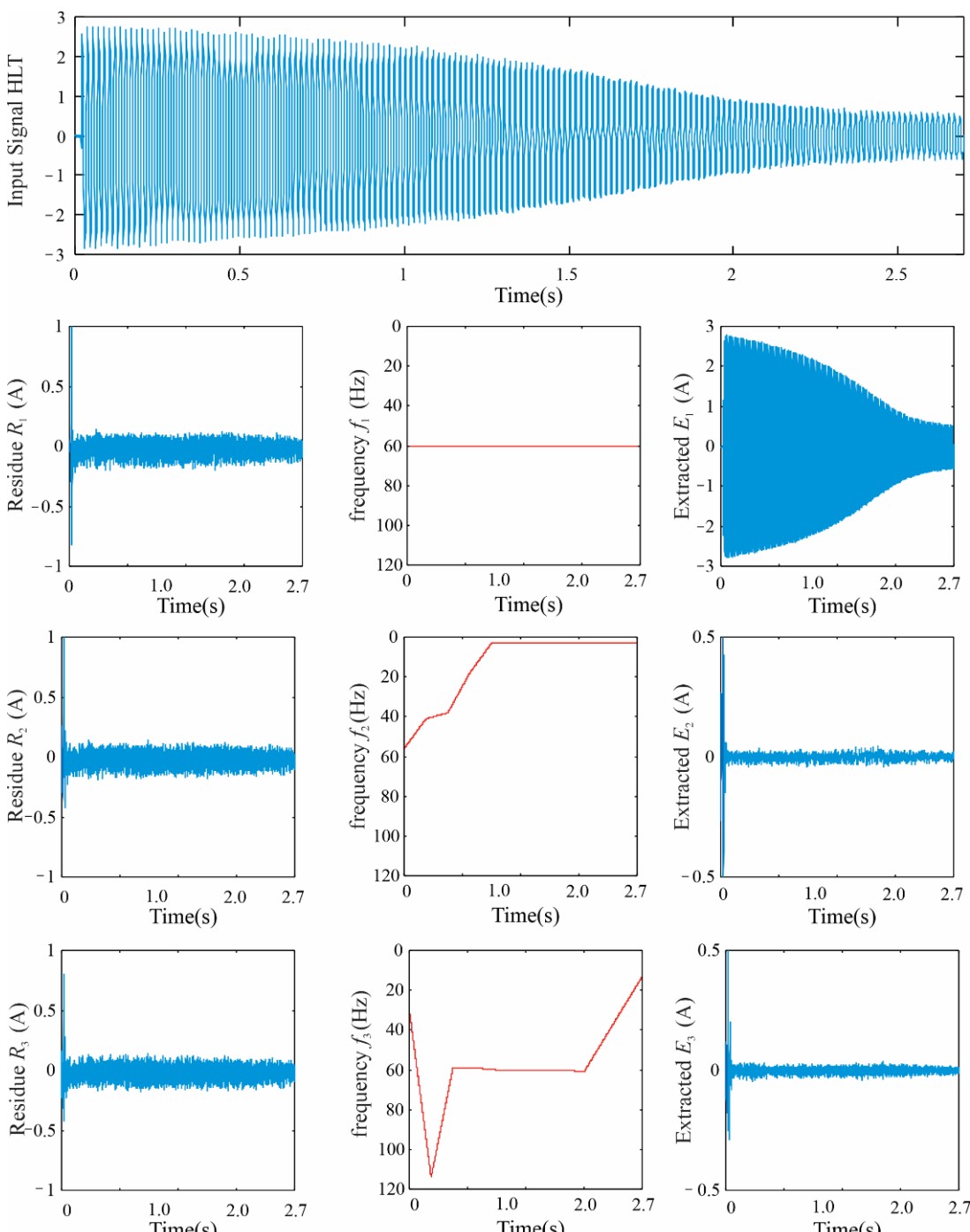

**Figure 10.** FBASD output for the HLT case.

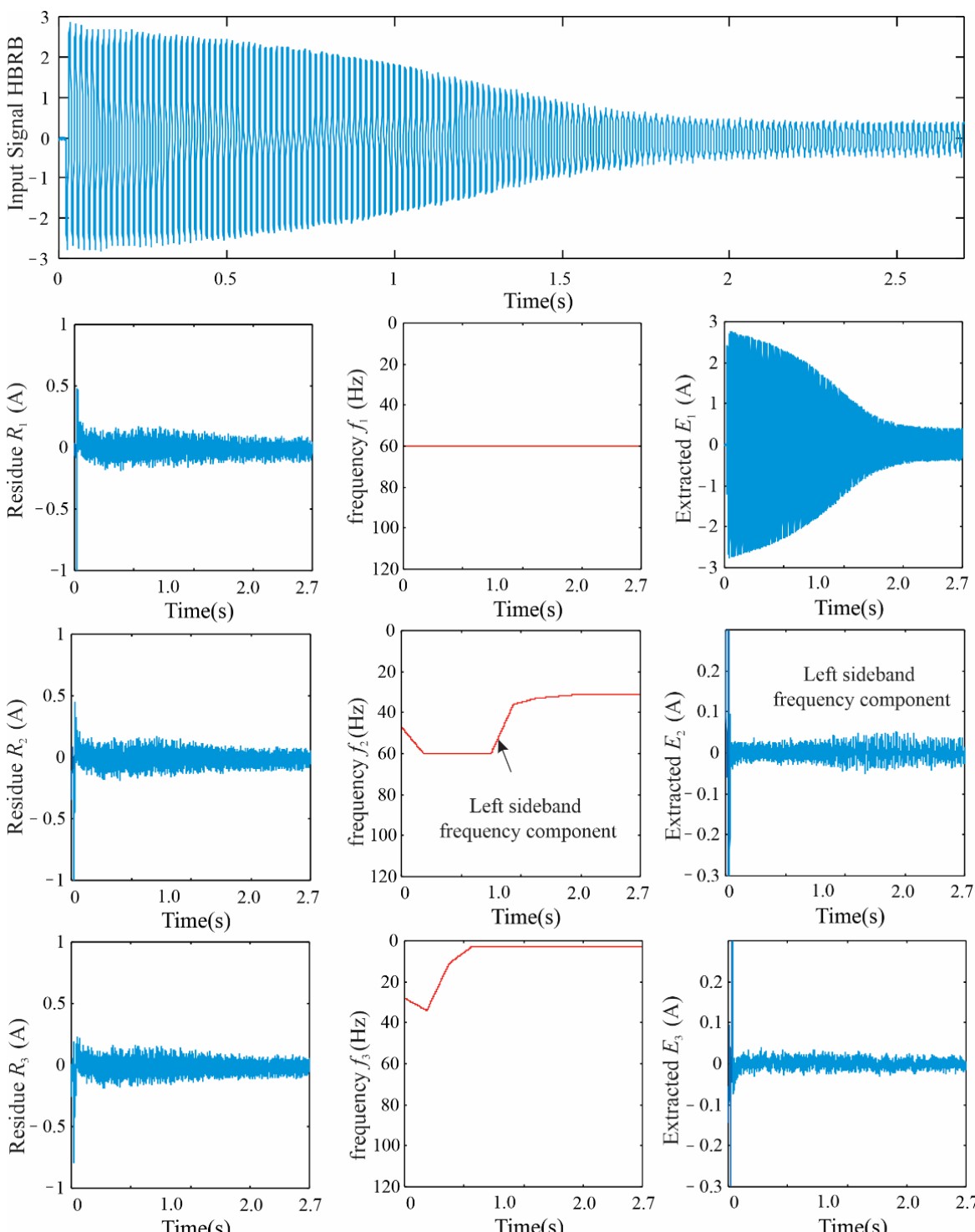

**Figure 11.** FBASD output for the HBRB case.

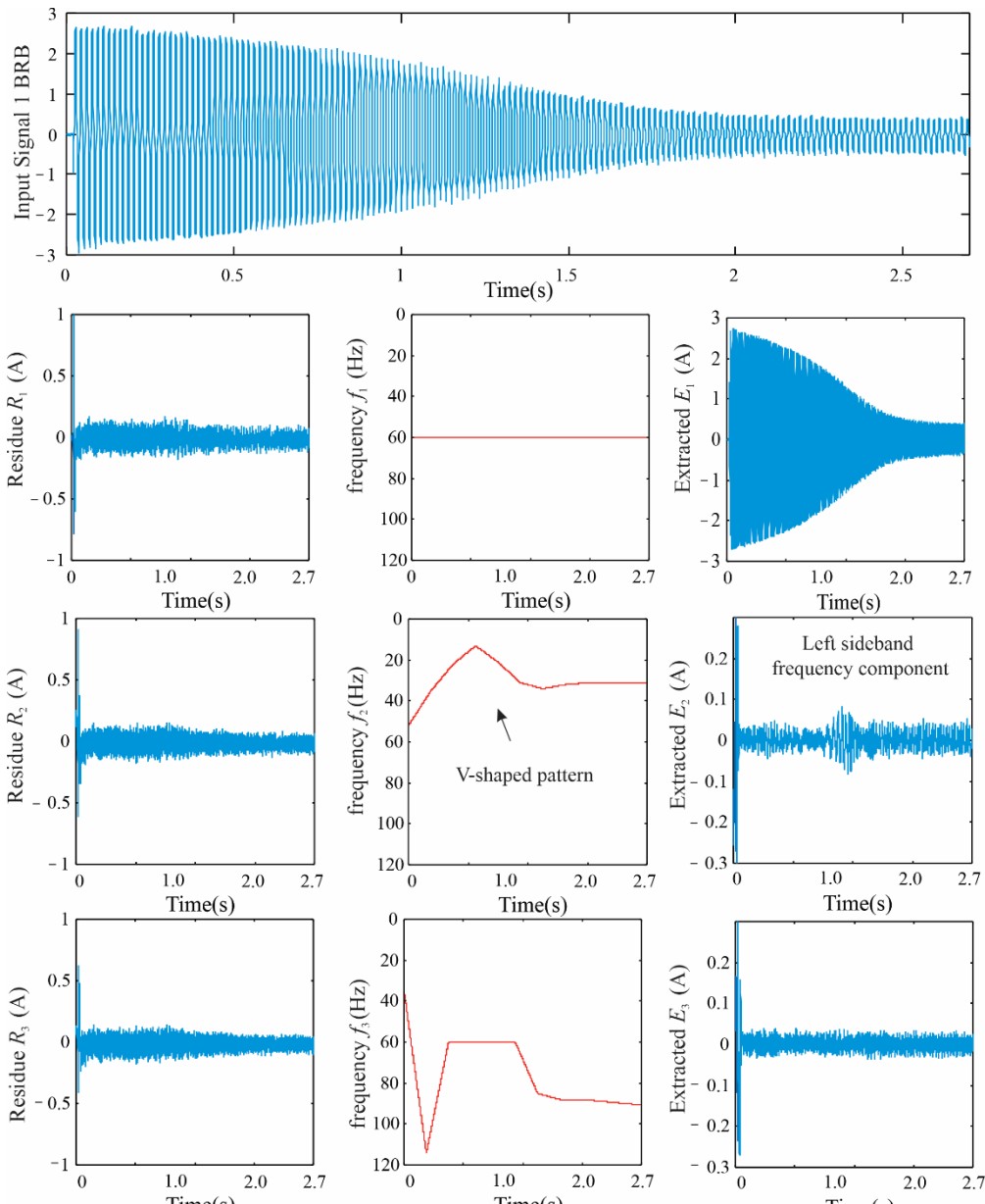

**Figure 12.** FBASD output for the 1BRB case.

Therefore, it is proposed to analyze the amplitude of the obtained signals components through the behavior of their corresponding envelopes since the components with greater amplitude will have upper envelopes with larger values. The envelope of a signal is a curve that limits its extreme values, and it is made up of the upper envelope and the lower envelope. Several techniques have been reported to calculate the envelope of a signal, e.g., the Hilbert or Wavelet transforms [41]. However, in this work, a very simple technique based on the quantization of the signal using its extreme values is proposed. In the case of the upper envelope, local maximum values are considered. An easy method to measure the amplitude of the envelope in a certain interval is calculating its energy in that interval, the total energy of a discrete signal $x[k]$ in an interval $1 \leq k \leq n$ is given by [35]:

$$E = \sum_{k=1}^{n} (x[k])^2 \tag{14}$$

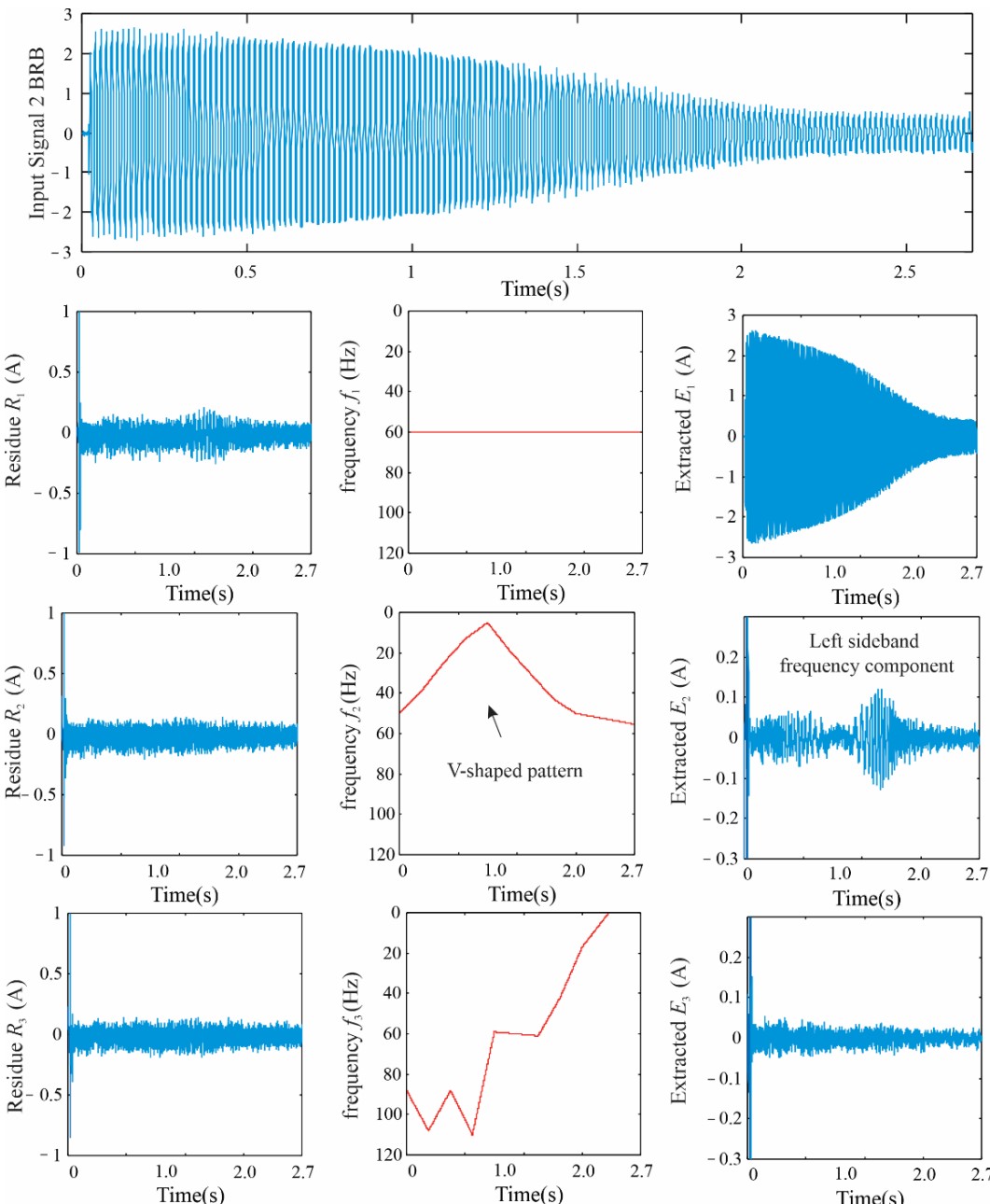

**Figure 13.** FBASD output for the 2BRB case.

In summary, the proposed signal classification algorithm is shown in Figure 14.

### 3.4. Classification Results

As mentioned in previous sections, a total of 80 tests were performed during the startup transient, i.e., 20 for each IM condition (HLT, HBRB, 1BRB, 2BRB). Next, according to the proposed FBASD algorithm, the second component of the signal in the time domain is extracted. Then the upper envelope is computed by a quantizing step where constant segments at the local maximum values are fixed. That is, all the local maximum or peak values in the component are searched and every maximum will remain as a constant value in the envelope signal until a new local maximum value is found, so the resulting envelope signal is a piecewise signal of constant values. In more detail, if the component $A$ has $m$ peak values located in the samples $k_1, k_2, k_3, \ldots, k_m$, then the upper envelope is a signal $x[k]$ of the same length as $A$ that satisfies $x[k] = A[k_i]$ for $k_{i-1} \le k < k_i$, $1 \le i \le m$, and

$k_0 = 1$ corresponds to the initial sample of $A$. Therefore, the amplitude of the constant segments is variable and is determined according to the waveform of the signal, that is, the envelope is defined according to the distribution of existing peak values in the signal. Figure 15 shows an example of the computed envelopes for each IM condition.

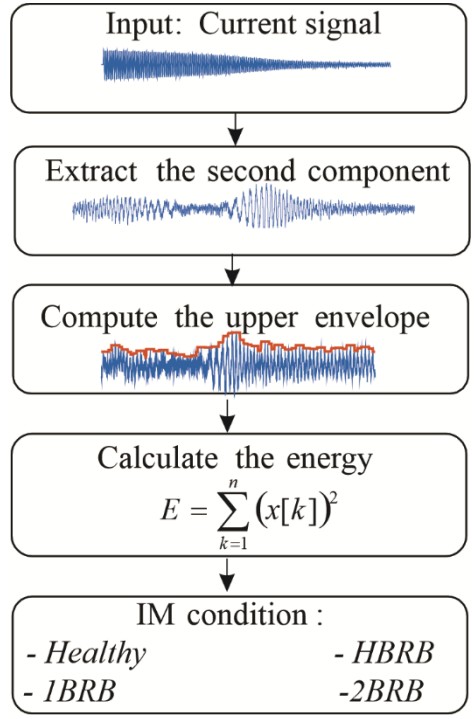

**Figure 14.** Proposed algorithm for the signal classification.

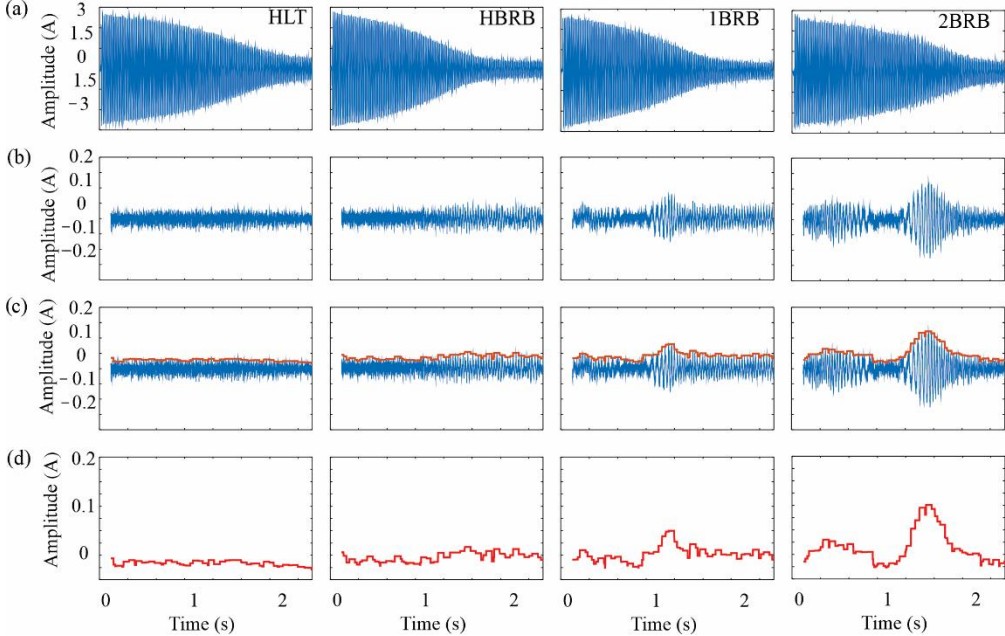

**Figure 15.** (**a**) Current signal, (**b**) Second extracted component, (**c**) Quantization of the component, (**d**) Upper envelope.

In Figure 15d, it can be seen that the amplitude of the component increases according to the degree of severity in the broken bars, as indicated by the background discussed. To quantitatively corroborate this phenomenon, the energy of the resulting envelopes is

calculated. The results of the calculated energy values for the 80 tests performed are shown graphically in Figure 16. Table 3 shows the corresponding mean and standard deviation values for each rotor condition.

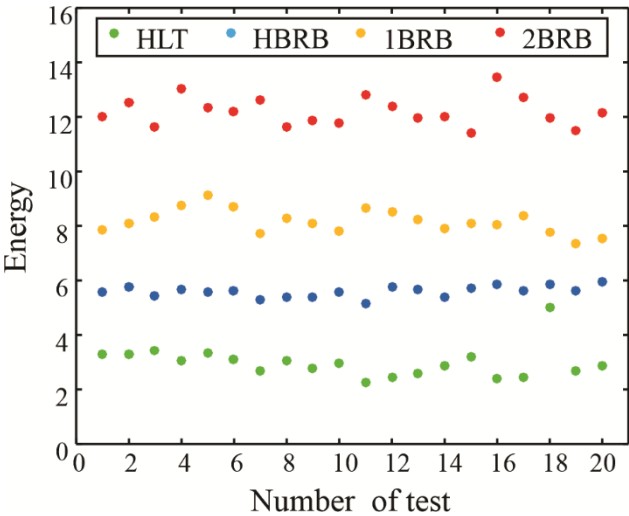

**Figure 16.** Energy calculated values.

**Table 3.** Mean and standard deviation of the energy values.

| IM Condition | Mean | Standard Deviation |
| --- | --- | --- |
| HLT | 2.98 | 0.586 |
| HBRB | 5.59 | 0.206 |
| 1BRB | 8.16 | 0.441 |
| 2BRB | 12.19 | 0.532 |

A one-way analysis of variance (ANOVA) was performed on the calculated energy data in order to validate its statistical significance among the rotor condition study cases. The ANOVA results are summarized in Table 4. A boxplot of the data is also presented in Figure 17. Where the calculated energy values can be used to distinguish between the different rotor conditions. According to the results of the experimentation, only one outlier was found, that is, an observation that presents an abnormal distance from other values, it is indicated in the graph with the red marker.

**Table 4.** Analysis of variance (ANOVA).

| Source | Sum of Squares | Degrees of Freedom | Mean Squared Error | F Statics | $p$-Value |
| --- | --- | --- | --- | --- | --- |
| Columns | 924.654 | 3 | 308.218 | 1425.52 | $1.1 \times 10^{-66}$ |
| Error | 16.432 | 76 | 0.216 | | |
| Total | 941.086 | 79 | | | |

The ANOVA presents a very low $p$-value, which determines that the calculated energy values of the envelopes are statistically different for each motor condition; therefore, this parameter can be used to discriminate the conditions of the analyzed IM. Additionally, the corresponding boxplot shows that the data groups are separated from each other, excepting possibly a test of the healthy condition that presents an outlier value of energy. Considering these results, a simple but effective if-then-else automatic classifier can be proposed. In order to do so the first five calculated energy values (25% of the tests) were taken in each condition and their average value was calculated to be the reference value which allows the classification stage. These values are shown in Table 5. Therefore, the classifier works as

follows: given a monitored current signal, the energy value is calculated with the previously mentioned methodology, then the differences of this value and the four average energy values shown in Table 5 are calculated and, finally, the value that presents the minimum difference will determine the motor condition. The performance of the proposed classifier was validated with the remaining 15 signals of each case study case, i.e., a total of 60 signals (75% of the tests) for validation. The classification results are summarized in the confusion matrix presented in Table 6, where an effectiveness higher than 93% is obtained in all the cases.

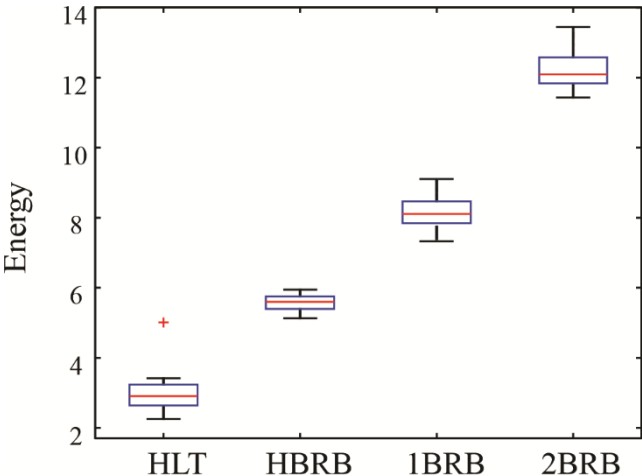

**Figure 17.** Boxplot of the calculated energy data.

**Table 5.** Mean energy value of the first five test.

|      | **HLT** | **HBRB** | **1BRB** | **2BRB** |
|------|---------|----------|----------|----------|
| Mean | 3.27    | 5.60     | 8.42     | 12.30    |

**Table 6.** Classification results (confusion matrix).

| IM Condition | HLT | HBRB | 1BRB | 2BRB | Effectiveness (%) |
|--------------|-----|------|------|------|-------------------|
| HLT  | 14 | 1  | 0  | 0  | 93.3 |
| HBRB | 0  | 15 | 0  | 0  | 100  |
| 1BRB | 0  | 0  | 15 | 0  | 100  |
| 2BRB | 0  | 0  | 0  | 15 | 100  |
|      | Total effectiveness | | | | 98.3 |

## 4. Discussion

Both the proposed FBASD algorithm and the signal classification stage applied to the detection of broken bars in induction motors present good results since 98.3% of total effectiveness was achieved in the validation tests, where only one test was not correctly classified. The success of the classifier stands out because of the FBASD stage since the main frequency components contained in the analyzed current signals can be clearly detected, where both stationary and transient frequency components are isolated, and their behavior is tracked during the monitoring time. Subsequently, the detected components can be extracted from the original signal to analyze them separately. This is a paramount feature of the proposed FBASD technique, since the adequate signal decomposition allows an additional analysis or processing that can be performed only on the selected time-frequency components where the negative effect of unwanted frequency components is minimized. In fact, the second extracted component in time and frequency domains showed that the amplitudes increased according to the degree of severity of the motor damage. This fact

had been already reported in the literature and it was corroborated with the tests carried out in this work.

## 5. Conclusions

A novel signal decomposition technique, based on an adaptive band-pass filter and STFT, was introduced in this work. The FBASD technique allows detecting and extracting frequency components embedded into a signal even if they vary in both amplitude and frequency. Also, the proposed technique is able to extract time-frequency components within a user-defined frequency range. The FBASD technique is validated through simulation and experimental study cases. In the simulation stage, the proposed technique is applied to a synthetic signal composed of three static frequency components and a quadratic chirp signal plus 10 dB of white Gaussian noise. The proposed FBASD method was demonstrated to be more robust to noise than the EMD and DWT techniques; moreover, it obtained the lowest computing time (from 5 to 300 times less time).

In the experimental study case, the FBASD technique was employed to decompose the startup-transient current signal obtained from an IM under different rotor fault conditions. The proposed FBASD technique was capable of isolating the time-frequency components of both the main power supply and the fault-related component. Once the extracted components were obtained, a classification algorithm was proposed for detecting HLT, HRBB, 1BRB, and 2BRB. The results showed a 98.3% of total effectiveness, where it is worth noting that a simple but effective energy-based classification algorithm was used.

Although the simulation and experimentation demonstrated the capabilities of the FBASD algorithm, the application of the proposal in other applications needs to take into consideration some possible drawbacks. For example, there is a balance between the time and frequency accuracy due to the inherent performance of the STFT. If the time accuracy is increased, the frequency accuracy will be decreased. A better time accuracy requires smaller windows size in Step 1 of the proposed technique. Subsequently, the noise will decrease the tracking of the frequency components. Additionally, since the proposed technique is based on the STFT, the leakage can decrease its performance in the presence of a big frequency component outside the user-defined frequency range. This effect could be mitigated if a band-pass filter is added as a signal preprocessing stage. Although a suitable method for proposing time and frequency windows is given in the proposed methodology, future work is required for fully adaptive windows determination. Furthermore, other interpolators based on polynomials or splines, and other features of the STFT method such as the overlap size and the window type (e.g., Hanning, Blackman, or others) to segment the analyzed signal will be explored, where a balance between the computational cost and their complexity for hardware implementation will be assessed.

**Author Contributions:** Conceptualization, J.J.D.S.-P. and J.R.R.-G.; methodology, J.J.D.S.-P., M.V.-R. and J.R.R.-G.; software, validation, and formal analysis, all authors; investigation, resources, and visualization, J.P.A.-S., G.I.P.-S. and M.T.-H.; data curation, J.J.D.S.-P. and J.P.A.-S.; writing—original draft preparation, writing—review and editing, all authors; supervision and project administration, M.V.-R. and J.R.R.-G.; funding acquisition, G.I.P.-S., M.T.-H. and J.R.R.-G. All authors have read and agreed to the published version of the manuscript.

**Funding:** This work was partially sponsored by FOFIUAQ-2018 and FIN-2019-04.

**Institutional Review Board Statement:** Not applicable.

**Informed Consent Statement:** Not applicable.

**Data Availability Statement:** The data presented in this study are not publicly available due to privacy issues.

**Conflicts of Interest:** The authors declare no conflict of interest.

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
