# Peer review of "Fourier-Based Adaptive Signal Decomposition Method Applied to Fault Detection in Induction Motors"

_machines, doi:10.3390/machines10090757_

Round 1

Reviewer 1 Report

In this paper, a Fourier-based adaptive signal decomposition method applied to fault detection in induction motors is proposed. This approach may be of great interest to readers; however, some points need to be clarified before a possible publication.

-How do you define the size of the XZ sale? Where do you get equation 1?

-Would the results be improved if a square or cubic interpolation is used? Address this issue further.

-What happens to BRB failure frequency harmonics?

-The same methodology can be applied instead of using the STFT only the Notch filter for sale WZ.

-Would the STFT overlap result in a better interpolation of fk?

-Would the proposal improve with the use using a different window than the rectangular one?

-How would the proposal respond to the phase changes of the current signal?

-Was there any cross-validation?

-Check table 5 "Title2, Title 3"

Reviewer 2 Report

The paper is well written and the methodology sounding.

Before publication, I suggest the Authors to add a discussion on the computational cost of the proposed methodology making a comparison with EMD and DWT.

Round 2

Reviewer 1 Report

The authors have improved the quality of their work according to the suggestions made. Hence, I can recommend its acceptance.